# Comparison of Different Near-Infrared Technologies to Detect Sentinel Lymph Node in Uterine Cancer: A Prospective Comparative Cohort Study

**DOI:** 10.3390/ijerph19127377

**Published:** 2022-06-16

**Authors:** Stefano Restaino, Nicolò Bizzarri, Vincenzo Tarantino, Silvia Pelligra, Rossana Moroni, Emilia Palmieri, Giorgia Monterossi, Barbara Costantini, Giovanni Scambia, Francesco Fanfani

**Affiliations:** 1Department of Obstetrics, Gynecology, and Pediatrics, Obstetrics and Gynecology Unit, Udine University Hospital, DAME, 33100 Udine, Italy; stefano.restaino@asufc.sanita.fvg.it; 2UOC Ginecologia Oncologica, Dipartimento per la Salute della Donna e del Bambino e della Salute Pubblica, Fondazione Policlinico Universitario A. Gemelli IRCCS, 00168 Rome, Italy; nicolo.bizzarri@yahoo.com (N.B.); silviasarap@gmail.com (S.P.); giorgia.monterossi@policlinicogemelli.it (G.M.); bacostantini@gmail.com (B.C.); giovanni.scambia@policlinicogemelli.it (G.S.); 3Università Cattolica del Sacro Cuore, 00168 Rome, Italy; vincenzo@tarantino.email (V.T.); emilia.palmieri01@icatt.it (E.P.); 4Fondazione Policlinico Universitario A. Gemelli IRCCS, Direzione Scientifica IRCCS, L.go Agostino Gemelli 8, 00168 Rome, Italy; roxinki@gmail.com

**Keywords:** endometrial cancer, sentinel lymph node, laparoscopy, near infrared, indocyanine green

## Abstract

Objectives: Sentinel lymph node biopsy is considered a crucial step in endometrial cancer staging. Cervical injection has become the most favored technique and indocyanine green has been demonstrated to be more accurate than other tracers. Different near-infrared camera systems are currently being used to detect indocyanine green in sentinel lymph nodes and have been compared in different patients. The present study aimed to determine the number and site of sentinel lymph nodes detected in the same patients with two different near-infrared technologies. Methods: This is a prospective, single-center, observational, non-sponsored study. Patients with presumed uterine-confined endometrial cancer were prospectively enrolled. After cervical injection, two different near-infrared cameras were used to detect sentinel lymph nodes at the same time: Olympus, Tokyo, Japan—considered the standard (SNIR); and Medtronic, Minneapolis, MN, USA with VISION SENSE^®^ which is a new laser near-infrared (LNIR) fluorescence laparoscope. The two cameras were alternatively switched on to detect sentinel lymph nodes in the same patients. Results: Seventy-four consecutive patients were included in the study. Most of the patients were diagnosed with endometrioid histology (62, 83.8%), FIGO stage IA (48, 64.9%), grade 2 (43, 58.1%), and underwent surgery with laparoscopic approach (70, 94.0%). The bilateral detection rate was 56/74 (75.7%) with SNIR and 63/74 (85.1%) with LNIR (*p* = 0.214). The total number of sentinel lymph nodes identified in the left hemipelvis was 65 and 70 with SNIR and LNIR, respectively; while in the right hemipelvis, there were 74 and 76, respectively. The median number of sentinel lymph nodes identified with SNIR and LNIR was 2 (range, 0–4) and 2 (range, 0–4), respectively (*p* = 0.370). No difference in site of sentinel lymph node detection was evident between the two technologies (*p* = 0.994). Twelve patients (16.2%) had sentinel lymph node metastasis: in all cases metastatic sentinel lymph nodes were detected both with Olympus and LNIR. Conclusions: No difference in bilateral detection rate and number or site of sentinel lymph node detection was evident comparing two different technologies of near-infrared camera for ICG detection in endometrial cancer patients. No difference in sentinel lymph node metastases identification was detected between the two technologies.

## 1. Introduction

Endometrial cancer is the most common gynecological tumor in developed countries and the second most common gynecologic cancer worldwide when both high- and low-income countries are considered [1]. Endometrial cancer develops in 1–3% of women; the average age at diagnosis is 63 years, the incidence peaks between ages 60 and 70 years, but 2–5% of cases occur before age 40. The number of people diagnosed with uterine cancer has increased by about 1% each year since the mid-2000; the increasing incidence of endometrial cancer may be potentially associated with several risk factors, including the growing prevalence of obesity, the prevalence of diabetes, and changes in reproductive behavior, diet, and exercise [2]. Most women with endometrial cancer initially present abnormal uterine bleeding due to shedding of the malignant endometrial lining, therefore endometrial cancer is mostly diagnosed at an early stage of disease given its symptomatic nature. In these stages, the standard treatment consists of total hysterectomy with bilateral salpingo-oophorectomy with lymph node assessment [3]. For many years, standard pelvic and para-aortic lymphadenectomy of the pelvic and paraaortic nodes was performed as part of the initial surgical evaluation, with the development of lymphedema in more than 30% of patients, causing discomfort, or heaviness and reduced mobility for some patients, and with short-term risks that included prolonged surgical times and increased blood loss [4,5]. “Sentinel” lymph node refers to one or several lymph nodes that are the primary landing zone of cancer cells that spread through the regional lymphatic drainage pathway; theoretically, if a sentinel lymph node is negative, lymphatic metastasis of the drainage area does not occur yet, confirming pN0 (no morphologic evidence of regional lymph node involvement in accordance with TNM classification system). Therefore, a sentinel lymph node could serve as a middle ground between comprehensive lymphadenectomy and no lymphadenectomy being associated with a substantially lower risk of post-operative morbidity even in gynecological cancer: many studies have proven sentinel node biopsy to stage endometrial cancer accurately so the international guidelines nowadays consider sentinel lymph nodes a valid option in presumed uterine-confined endometrial cancer staging [6,7]. For these reasons, in the past few decades sentinel lymph node detection has become routinely performed. Cervical injection has become the most favored technique since the cervix is easily accessible, and the parametrial lymphatic drainage has proved to be the main route of drainage of the uterus. Indocyanine green has been demonstrated to be more accurate than other tracers, such as technetium and blue dye, in sentinel lymph node detection: when injected outside blood vessels, indocyanine green binds to proteins and is found in the lymph, reaching the nearest draining lymph node usually within 15 min; it becomes fluorescent once excited, either using a laser beam or near infra-red light and can be detected using specifically designated scopes and cameras [8,9,10]; moreover, indocyanine green’s safety profile, ease of use and effectiveness in identifying lymph nodes may favor its employment with respect to conventional tracers. A randomized study comparing indocyanine green with blue dye showed that indocyanine green identified more sentinel nodes than isosulfan blue dye, with no difference in the pathological confirmation of nodal tissue between the two mapping substances [9]. The advent of near-infrared fluorescent (NIR) technology in various laparoscopic and robotic platforms paved the way for developing new products to detect sentinel lymph nodes: several industry device manufacturers are actively developing novel imaging technologies with enhanced computer software to produce more precise and reliable fluorescent imaging [10,11,12,13,14]. A recent system, the Medtronic^®^ Elevision IR camera system, has been developed, and it uses innovative laser technology in conjunction with indocyanine green for high-definition imaging and produces simultaneous white light and infrared (IR) fluorescence images. This camera merges the two images in real-time, providing a real-time qualitative and quantitative measurement of IR signal intensity and creating a uniform edge-to-edge illumination pattern in procedures, resulting in sharp peripheral images and reliable measurements [15].

Moreover, there is increasing concern about the phenomenon of ‘empty node packet’: a sentinel lymph node specimen without evidence of lymph node(s) is defined as an empty node; this event seems to be one of the main disadvantages of using indocyanine green due to its rapid extravasation when the surgeon starts the dissection [16]. Bedyńska et al. reported that almost 40% of the detection failure with ICG was due to ‘empty packets’ [17]. Using color-based software that coordinates with different levels of indocyanine green uptake, the camera may help surgeons to distinguish the “true” sentinel lymph node from fatty tissue that, although absorbing fluorescent dye, does not contain true nodal tissue [16,18] and from non-sentinel lymph nodes which may be fluorescent, avoiding empty nodes collection. Therefore, the Medtronic^®^ camera may allow the detection of higher lymph nodal tissue quality as well as the “true” sentinel lymph nodes. Since a recent study showed that a higher sentinel lymph node count does not seem to increase the accuracy of sentinel lymph node mapping in cervical and endometrial cancer patients [12] and many studies proved how the incidence of lymphedema is related to the number of lymph nodes removed, this device, detecting the “true” sentinel lymph nodes, may allow for more precise surgery thus avoiding unnecessary surgical trauma.

The uterus has a complex lymphatic drainage: there are two main interconnected routes through broad and infundibulo-pelvic ligaments; these last ones are considered the secondary routes. Studies identified different drainage routes for the different parts of the organ: the cervix and the lower uterine segment usually drain through parametria to external iliac and common iliac nodes. The uterine corpus drains primarily to the external iliac nodes; other pathways include the inter iliac, common iliac, and obturator area. Uterine fundal drainage is relatively consistent through the ovarian vessels to the infra-renal paraaortic area [19].

The present study aimed to evaluate the bilateral detection rate compared between the two systems in determining the number of sentinel lymph nodes detected in the same patient. A secondary aim was to assess the sentinel lymph node metastasis identified with the two systems.

## 2. Materials and Methods

This is a prospective, single-institution, observational, non-sponsored study. Between September 2020 and May 2021, 74 consecutive patients who were surgically staged for primary treatment of endometrial cancer were enrolled at Fondazione Policlinico Universitario Agostino Gemelli, IRCCS, in Rome, Italy. We included all women undergoing sentinel lymph node biopsy, in accordance with international guidelines. As per international guidelines, all sentinel lymph nodes were sent for ultrastaging at final pathology [3]. We excluded from the study patients with allergy to indocyanine green or iodine, patients not eligible for surgery, or patients with body mass index (BMI) > 35 since in our hospital these patients are destined for a robotic approach. To avoid any falsification of the lymphatic drainage, we ruled out patients with positive anamnesis of previous surgery to the pelvis since the formation of adhesion in the retroperitoneum may alter lymphatic pathways and mapping. The primary objective was to evaluate the inter-rater reliability between two different fluorescence system technologies in determining the number of sentinel lymph nodes in the same patient. Secondarily, we investigated the number of sentinel lymph nodes visually detected intra-operatively with the two different systems, with final histology. The study protocol was approved by the Ethics Committee (protocol number 0035693/20, ID 3340). All patients were adequately informed and inserted in the study only after having read and signed the specific informed consent. Diagnostic, clinical, and surgical data of each patient were prospectively recorded. At the end of the procedure, a case report form was compiled with intraoperative data. All clinical and histologic data were reported on an anonymous electronic database. Post-operative complications will be evaluated during the first 30 days after surgery according to Dindo’s classification. 

### 2.1. Surgical Procedures

While under general anesthesia, the patient was positioned in the dorsal lithotomic position with both legs supported in stirrups with a Trendelenburg tilt and arms along the body. A four disposable or reusable, sterile trocar transperitoneal approach was used. A 12 mm port was inserted at the umbilicus for the telescope. Once pneumoperitoneum (12 mmHg) was achieved, intra-abdominal visualization was obtained with a 0° high-definition telescope (Olympus Winter & IBE GMBH, Hamburg, Germany) considered the standard (SNIR) or with 0° Visionsense™ VS3 Iridium scope (Medtronic Italia SpA), which is a new laser near-infrared (LNIR) fluorescence laparoscope for bright-field full-color observation and provides the ability to adjust the intensity of excitation light and quantify the intensity of indocyanine green fluorescence during observation [14,15].

Two additional 5 mm ports were placed under direct vision, in the right and left lower abdomen. One further 5-mm trocar was inserted in the suprapubic area. 

Indocyanine green dye powder was diluted in 10 mL sterile water obtaining a final solution of 2.5 mg/mL of fluorescent dye. Overall, 2 mL of the indocyanine green solution was injected into the cervix in the operating theatre, after the induction of general anesthesia and once the operative trocars were inserted. For sentinel lymph node mapping in our procedures, we followed a well-defined mapping algorithm proposed by Barlin et al. of the Memorial Sloan-Kettering Cancer Center; an algorithm that showed high sensitivity [20]. The injection was performed 0.5 mL superficial (1.3 mm) and deep (1–2 cm) in the cervix, at 3 and 9 o’clock slowly to maximize lymphatic uptake and minimize staining of deep pelvic tissues. The round ligaments were coagulated and cut to enter the retroperitoneum. With Medtronic’s camera, after about 15 min it was possible to visualize the lymphatic course, even without access to the retroperitoneum. Instead, within 20 min of waiting it was possible to see the course of the lymphatic system with our standard camera (Olympus camera). The two cameras were alternatively switched on to detect sentinel lymph nodes in the same patients. In this way, we compared, in the same patient, if the two cameras identified the same “true” sentinel lymph nodes. Moreover, the quantitative colorimetric evaluation of Medtronic’s was recorded with Medtronic’s camera. 

We identified the sentinel lymph nodes, evaluating each hemipelvis individually, with the two cameras, alternately. The sentinel lymph nodes were identified transperitoneally first and after retroperitoneal space development after. The sentinel lymph nodes were then retrieved as previously described [21]. If sentinel node mapping did not occur, pelvic lymphadenectomy was performed, and all the lymph nodes were removed via endobag. We also removed any suspicious or enlarged lymph nodes. Pelvic lymph nodes were sent for the ultrastaging. At the conclusion of the surgical procedure, the abdomen was deflated before removing the trocars.

### 2.2. Statistical Analysis

Sample size was calculated based on the primary objective of evaluating the disagreement rate between two different fluorescence system technologies in determining the number of SLN detected in the same patient. Hence, assuming a disagreement rate of 5%, a level of confidence of 0.05 and a power of 90%, the required sample size was N = 73. The sample was described in its clinical and demographic characteristics using descriptive statistics techniques. Qualitative variables were summarized by their absolute and percentage frequencies. Quantitative variables are presented by their mean and standard deviation (or median and range, when appropriate). The primary objective was achieved, calculating Cohen’s Kappa index along with its 95% confidential interval for the two near-infrared technologies. The description of the site of indocyanine green was performed with descriptive statistics. The comparison of the number of sentinel lymph nodes visually detected with the two different systems and the histology (gold standard) were performed with Cohen’s Kappa index. A *p*-value < 0.05 was considered statistically significant (2-tailed test). Statistical analysis was performed using SPSS statistical software, version 20 (SPSS Inc., Chicago, IL, USA).

## 3. Results

A total of N = 74 patients (median age of 60 years, with a range from 29 to 75) were enrolled in this study. 

The median age of patients was 60 years (range, 29–75). The mean body mass index of patients was 24.1 kg/m^2^ (range, 18.3–34.9). Sixty-four (79.7%) patients with endometrial cancer presented were diagnosed with endometrioid histology; for five patients, preoperative histotype was not available. Forty-one (55.4%) patients presented with an apparent IA FIGO stage, 25 (33.7%) IB, 17 (22.9%) II, 1 (1.4%) IIIA; for 12 patients (16.2%) preoperative imaging did not specify myometrial or serosa invasion. All sentinel lymph nodes were detected by a minimally invasive laparoscopic approach. The median duration of the entire procedure was 119 min (range 45–618) with a median estimated blood loss of 50 mL (range 20–300). After final pathology, 62 tumors (83.8%) were classified as endometrioid tumors, while 12 as non-endometrioid. Forty-eight cases (64.9%) were staged as FIGO IA, 11 (14.9%) as FIGO IB, three (4.1%) as FIGO II, one (1.4%) as FIGO IIIA, one (1.4%) as FIGO IIIB, six (8.1%) as FIGO IIIC1, one (1.4%) as FIGO IIIC2, and three (4.1%) as FIGO IVB. The median diameter of tumors was 25.5 mm (range 3–110 mm); 27 (36.5%) presented lymphovascular space invasion and four (5.5%) showed a MELF pattern. All patients underwent sentinel lymph node mapping, 24 (32.4%) received a second injection since the first one was not successful, re-injection was unilateral in 19 (25.7%) women and bilateral in five (6.7%). Using the Medtronic camera (LNIR) for lymph node mapping, 63 (85.1%) had a bilateral mapping. Among the other 11 (14.9%) patients, nine (12.2%) had unilateral mapping and two (2.7%) had no mapping. Using the Olympus camera (SNIR) instead, 56 (75.7%) patients had a bilateral mapping, 16 (21.6%) had unilateral mapping and two (2.7%) had no mapping. LNIR therefore showed a trend towards a better bilateral detection rate; this finding, however, did not reach statistical significance (*p* = 0.214). The total number of sentinel lymph nodes identified on the left hemipelvis was 65 and 70 with SNIR and LNIR, respectively; while in the right hemipelvis, it was 74 and 76, respectively. In 13 patients (17.6%) a side-specific pelvic lymphadenectomy was performed and in four (5.4%) a bilateral pelvic lymphadenectomy. Two (2.7%) patients underwent both pelvic lymphadenectomy and paraaortic (Table 1). The median number of sentinel lymph nodes identified with SNIR and LNIR was two (range, 0–4) and two (range, 0–4), respectively (*p* = 0.370). No difference in the site of sentinel lymph nodes detection was evident between the two technologies (*p* = 0.994). Twelve patients (16.2%) had sentinel lymph nodes metastasis at final pathology: in all cases sentinel lymph nodes were detected both with Olympus and LNIR. Nine (6.2%) out of 146 lymph nodes removed were positive for isolated tumor cells, one (0.7%) was positive for micrometastasis and six (4.1%) for macrometastasis: also here, in all cases, sentinel lymph nodes were detected with both Olympus and LNIR (Table 2).

In our study using SNIR we identified, in the right hemipelvis, in 31 patients (41.9%) an obturator sentinel lymph node, in 29 patients (39.2%) an external iliac sentinel lymph node, in four patients (5.4%) an internal iliac sentinel lymph node, in one patient (1.4%) a common iliac sentinel lymph node, in one patient a presacral lymph node and in two patients (2.7%) a paraaortic lymph node. In the left hemipelvis, using SNIR, we identified in 19 patients (25.7%) an obturator sentinel lymph node, in 33 patients (44.6%) an external iliac sentinel lymph node, in two patients (2.7%) an internal iliac sentinel lymph node, in five patients (6.8%) a common iliac sentinel lymph node and in one patient (1.4%) a presacral lymph node.

Using LNIR in the right hemipelvis, we identified in 32 patients (43.2%) an obturator sentinel lymph node, in 29 patients (39.2%) an external iliac sentinel lymph node, in four patients (5.4%) an internal iliac sentinel lymph node, in one patient (1.4%) a common iliac sentinel lymph node, in one patient a presacral lymph node and in two patients (2.7%) a paraaortic lymph node. 

In the left hemipelvis, using LNIR, we identified in 22 patients (29.7%) an obturator sentinel lymph node, in 35 patients (47.3%) an external iliac sentinel lymph node, in two patients (2.7%) an internal iliac sentinel lymph node, in five patients (6.8%) a common iliac sentinel lymph node and in one patient (1.4%) a presacral lymph node. 

We observed no post-operative complications during the first 30 days after surgery according to Dindo’s classification. 

In Table 3, Cohen’s Kappa Index results are presented along with the 95% Confidence Interval. Cohen’s Kappa has been performed on the following variables. In detail, we compare the number and the site of sentinel lymph nodes visually detected intra-operatively with the two different systems.

## 4. Discussion and Conclusions

Sentinel lymph nodes can be identified through different lymphatic pathways and in cases where more pathways are highlighted in the same patient, it is difficult to discriminate which are the “true” sentinel lymph nodes [22]. In this prospective, observational, cohort study we compared two platforms for sentinel lymph node mapping for endometrial cancers on the same patient. Technological innovations and the use of minimally invasive techniques allow for an accurate diagnosis and personalized treatment [23,24]. Different near-infrared cameras to detect indocyanine green sentinel lymph nodes were proposed [13,14,15]. A more recent system has been developed and it uses an innovative laser technology in conjunction with indocyanine green for high-definition imaging, produces simultaneous white light and infrared (IR) fluorescence images. It merges the two in real time, provides real-time qualitative and quantitative measurement of IR signal intensity, and creates a uniform edge-to-edge illumination pattern, resulting in sharp peripheral image and reliable measurements [13,14,15,25]. Furthermore, this camera allows us to differentiate different indocyanine green uptake in lymph nodes or lymphatic channels, thus highlighting the sentinel lymph node with maximal uptake. There are different articles in the literature comparing different tracers and injection routes for sentinel node detection, but few studies compare different visual systems [11,26]. Previously, Buda et al. reported the comparison of two different fluorescence systems in different patients. The authors concluded that both fluorescence systems were valid and applicable for sentinel lymph node detection in the case of early-stage cervical or endometrial cancer [27]. The main strength of this study is that two different cameras are used in the same patient. The main drawback of the study is represented by the relatively small number of patients included, which was however calculated before starting the recruitment. Probably the power of statistical comparison is limiting us to draw significant conclusions. With the present study, we showed that the two systems showed no difference in bilateral detection rate and in sentinel lymph node metastasis identification. Therefore, both near-infrared fluorescence indocyanine green systems are to be considered valid in the detection of sentinel lymph node mapping in endometrial cancer. Moreover, SNIR highlighted nodes that, at final pathology, were described as fibro adipose tissue—this confirms the potential of this system for detecting higher lymph nodal tissue quality as opposed to the fatty tissue that absorbs the fluorescence and can be confused with lymphatic tissue, avoiding “empty nodes” packets. LNIR seemed to allow a better lymphatic mapping and a better bilateral detection rate even if this finding did not reach statistical significance (*p* = 0.214). Another limitation was the fact that we could not measure the time between tracer injection and sentinel lymph node identification as we had to alternate the two cameras during sentinel lymph node identification. Furthermore, this camera allows us to differentiate different indocyanine green uptake in lymph nodes or lymphatic channels, thus highlighting the sentinel lymph node with maximal uptake.

In conclusion, our study showed no difference in bilateral detection rate and in number of sentinel lymph node metastasis identified with two different near-infrared cameras on the same patients. Further and larger studies should be performed in order to evaluate new technologies that aim to increase the bilateral detection rate and to identify “true” sentinel lymph nodes from fatty tissue that, although absorbing fluorescent dye, does not contain true nodal tissue.

## Figures and Tables

**Table 1 ijerph-19-07377-t001:** Characteristics of the patients.

Variables	N = 74 (Range, %)
Demographic and preoperative variables	
Median Age, years (range)	60 (29–75)
Median BMI, kg/m^2^ (range)	24.1 (18.3–34.9)
Preoperative histology	
Endometrioid	59 (79.7)
Serous	4 (5.4)
Clear-cell	3 (4.1)
Other	3 (4.1)
n.a.	5 (6.7)
Preoperative grading	
1	20 (27.0)
2	25 (33.7)
3	17 (22.9)
n.a.	12 (16.2)
Preoperative clinical FIGO stage	
IA	41 (55.4)
IB	18 (24.3)
II	2 (2.7)
IIIA	1 (1.4)
n.a.	12 (16.2)
Surgical Approach	
Laparoscopy	70 (94.0)
Laparotomy	4 (6)
Surgical procedures	
Total hysterectomy, bilateral salpingo oophorectomy	74 (100.0)
SLNB	74 (100.0)
Re-injection	24 (32.4)
Monolateral	19
Bilateral	5
Unilateral pelvic lymphadenectomy	13 (17.6)
Bilateral pelvic lymphadenectomy	4 (5.4)
Paraaortic lymphadenectomy	2 (2.7)
Operative time (min) [median]	119 (45–618)
Intraoperative bleeding (ml) [median]	50 (20–300)
Final histology	
Endometrioid	62 (83.8)
Non-Endometroid	12 (16.2)
Tumor characteristics	
Median diameter, mm (range)	25.5 (3–110)
LVSI	27 (36.5)
MELF pattern	4 (5.5)
Histologic grade	
1	11 (14.8)
2	43 (58.1)
3	19 (25.7)
n.a.	1 (1.4)
FIGO Stage	
IA	48 (64.9)
IB	11 (14.9)
II	3 (4.1)
IIIA	1 (1.4)
IIIB	1 (1.4)
IIIC1	6 (8.1)
IIIC2	1 (1.4)
IVB	3 (4.1)

Abbreviations: BMI: Body mass index. NA: Not applicable. SLNB: Sentinel lymph node biopsy. FIGO: International Federation of Gynecology and Obstetrics. MELF pattern: Microcystic, elongated, and fragmented pattern of invasion. LVSI: Lymph-vascular space invasion.

**Table 2 ijerph-19-07377-t002:** Number of patients identified with lymph nodes with the two different probes.

	SNIR	LNIR
No Mapping	2 (2.7%)	2 (2.7%)
Unilateral mapping	16 (21.6%)	9 (12.2%)
Bilateral mapping	56 (75.7%)	63 (85.1%)

**Table 3 ijerph-19-07377-t003:** Inter-rater reliability for the two near-infrared technologies.

Olympus vs. Medtronic—Cohen’s Kappa and 95% CI			
	Cohen’s Kappa	Inf 95% CI	Sup 95% CI
Mapping	0.722	0.53	0.91
SLN location right	0.980	0.94	1.02
SLN location left	0.886	0.80	0.93
SLN number right	0.834	0.68	0.91
SLN number left	0.654	0.44	0.76

## Data Availability

The data presented in this study are available on request from the corresponding author.

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
