# Peer review of "Comparison of Different Near-Infrared Technologies to Detect Sentinel Lymph Node in Uterine Cancer: A Prospective Comparative Cohort Study"

_ijerph, 2022, doi:10.3390/ijerph19127377_

Round 1

Reviewer 1 Report

The author revised the paper according to my previous comments

Reviewer 2 Report

Can be accepted in the current form.

This manuscript is a resubmission of an earlier submission. The following is a list of the peer review reports and author responses from that submission.

Round 1

Reviewer 1 Report

Reviewer comments:
Comments to the Author
The present article discusses different near-infrared camera systems to detect ICG sentinel lymph node in 
different patients. Authors aimed to determine the number of sentinel lymph node detected in the same 
patients with two different near-infrared technologies. This article’s introduction section is written very 
well and discussed appropriately while the data obtained need to be presented in tables and/ or bar graphs 
forms. However, authors are advised to incorporate some suggestions.
Major criticisms
• Please elaborate “pN0”, “SLN” etc., wherever first used in the manuscript.
• No table is provided as “Table 1”. Please provide the table.
• It is very difficult to conclude the outcomes which are discussed in the Result section. If the data 
obtained can be organized in terms of tables and bar graphs, then it will be easier to compare 
directly which technology is better.
• Please provide table showing the percent with the number of patients identified with lymphnodes.

Author Response

Thank you for your comments and the time you dedicated to our work. 

Here it is a new version of the manuscript. 

We corrected some minor language issue and also added two tables for a easier and faster understanding

Reviewer 2 Report

I have carefully read the paper entitled as "Comparison of different near-infrared technologies to detect 2 sentinel lymph node in uterine cancer: a prospective comparative cohort study" by Restaino et al. It is a well designed and well written study but I have several comments:

1) Authors summarized their results in the text but it would be better to summarize them in tables to better and easier understand the results. They have to make a table about the patients characteristics, another one about the tumor characteristics and another comparing the two methods' results.

2) The discussion section has to be enriched to better discuss the background of two methods and better understand for the people who is not very familiar with these methods.

3)Minor spelling and grammar check is required

Author Response

Thanks for your comments to our work. 

Please find attached a new implemented version of the manuscript. We added two tables to better understand the results.

Reviewer 3 Report

Review report
Paper title: Comparison of different near-infrared technologies to detect 2 sentinel lymph
node in uterine cancer: a prospective comparative cohort study.

Authors: Stefano Restaino at. al.

The paper is very interesting and have a potential to publish at the consider journal of MDPI
after the authors will address the following Major comments:

- The abstract is too short and need to be extended. The abstract should summarize the
methods and the results of the paper.

- The data in the introduction section need a list of references: lines 47 and so on. The
list of references is very poor and not wide and there is a lot of data in this section.

- The Statistical analysis section is very very poor. The authors must extend this
section. As well as the method section.

- There is no neat table of study parameters or data. The methodology is not well
explained. Data display and data usage are unclear. How the researchers used the
statistical methods presented for use in the study. Also, the methods are not clearly
explained to the reader. All of these must be clear and well explained to the reader in
order to enhance the article to an academic research level.

- The following paper can be added to the paper:

Analysis of a breast cancer mathematical model by a new method to find an optimal
protocol for HER2-positive cancer

OP Nave, M Elbaz, S Bunimovich-Mendrazitsky Biosystems

Author Response

Thanks for your comments and for the time you dedicated to our work. 

Please find attached a new versione of the manuscript. 

We added some lines to the abstract and the statistical section, we implemented the lists of references, and we added two tables to better understand the results. 

Hope you find this version more suitable for the journal. 

Round 2

Reviewer 1 Report

The manuscript looks better in the present format to be accepted.

Reviewer 3 Report

I can't see that the authors address all of my comments. For example, The abstract is still very short and does not summarize the paper. 

The list of references is very poor and not wide and there
is a lot of data in this section.

The Statistical analysis section is very very poor. The authors must
extend this section. As well as the method section.

The methodology is
not well explained. Data display and data usage are unclear. How the
researchers used the statistical methods presented for use in the study.
Also, the methods are not clearly explained to the reader. All of these
must be clear and well explained to the reader in order to enhance the
article to an academic research level.